# The Use of Nitrosative Stress Molecules as Potential Diagnostic Biomarkers in Multiple Sclerosis

**DOI:** 10.3390/ijms25020787

**Published:** 2024-01-08

**Authors:** Saskia Räuber, Moritz Förster, Julia Schüller, Alice Willison, Kristin S. Golombeck, Christina B. Schroeter, Menekse Oeztuerk, Robin Jansen, Niklas Huntemann, Christopher Nelke, Melanie Korsen, Katinka Fischer, Ruth Kerkhoff, Yana Leven, Patricia Kirschner, Tristan Kölsche, Petyo Nikolov, Mohammed Mehsin, Gelenar Marae, Alma Kokott, Duygu Pul, Julius Schulten, Niklas Vogel, Jens Ingwersen, Tobias Ruck, Marc Pawlitzki, Sven G. Meuth, Nico Melzer, David Kremer

**Affiliations:** 1Department of Neurology, Medical Faculty and University Hospital Düsseldorf, Heinrich Heine University Düsseldorf, 40225 Düsseldorf, Germany; saskiajanina.raeuber@med.uni-duesseldorf.de (S.R.); moritz.foerster@mariahilf.de (M.F.); j.schueller@hhu.de (J.S.); alicegrizzel.willison@med.uni-duesseldorf.de (A.W.); kristin.golombeck@med.uni-duesseldorf.de (K.S.G.); christinabarbara.schroeter@med.uni-duesseldorf.de (C.B.S.); menekse.oeztuerk@med.uni-duesseldorf.de (M.O.); robin.jansen@med.uni-duesseldorf.de (R.J.); niklas.huntemann@med.uni-duesseldorf.de (N.H.); christopher.nelke@med.uni-duesseldorf.de (C.N.); melanie.korsen@med.uni-duesseldorf.de (M.K.); katinka.fischer@hhu.de (K.F.); ruth.kerkhoff@mariahilf.de (R.K.); yana.leven@med.uni-duesseldorf.de (Y.L.); patricia.kirschner@med.uni-duesseldorf.de (P.K.); tristan.koelsche@med.uni-duesseldorf.de (T.K.); petyo.nikolov@med.uni-duesseldorf.de (P.N.); juliagelenar.marae@med.uni-duesseldorf.de (G.M.); alma.kokott@med.uni-duesseldorf.de (A.K.); duygu.pul@med.uni-duesseldorf.de (D.P.); julius.schulten@med.uni-duesseldorf.de (J.S.); niklas.vogel@med.uni-duesseldorf.de (N.V.); jens.ingwersen@med.uni-duesseldorf.de (J.I.); tobias.ruck@med.uni-duesseldorf.de (T.R.); marc.pawlitzki@med.uni-duesseldorf.de (M.P.); svenguenther.meuth@med.uni-duesseldorf.de (S.G.M.); nico.melzer@med.uni-duesseldorf.de (N.M.); 2Department of Neurology, Kliniken Maria Hilf GmbH, Academic Teaching Hospital of the RWTH Aachen University Hospital, 41063 Moenchengladbach, Germany; 3Department of Neurology and Neurorehabilitation, Hospital Zum Heiligen Geist, Academic Teaching Hospital of the Heinrich Heine University Düsseldorf, 47906 Kempen, Germany

**Keywords:** biomarkers, nitrosative stress molecules, relapsing remitting multiple sclerosis, primary progressive multiple sclerosis

## Abstract

Multiple sclerosis (MS) is an autoimmune disease of the central nervous system (CNS) of still unclear etiology. In recent years, the search for biomarkers facilitating its diagnosis, prognosis, therapy response, and other parameters has gained increasing attention. In this regard, in a previous meta-analysis comprising 22 studies, we found that MS is associated with higher nitrite/nitrate (NOx) levels in the cerebrospinal fluid (CSF) compared to patients with non-inflammatory other neurological diseases (NIOND). However, many of the included studies did not distinguish between the different clinical subtypes of MS, included pre-treated patients, and inclusion criteria varied. As a follow-up to our meta-analysis, we therefore aimed to analyze the serum and CSF NOx levels in clinically well-defined cohorts of treatment-naïve MS patients compared to patients with somatic symptom disorder. To this end, we analyzed the serum and CSF levels of NOx in 117 patients (71 relapsing–remitting (RR) MS, 16 primary progressive (PP) MS, and 30 somatic symptom disorder). We found that RRMS and PPMS patients had higher serum NOx levels compared to somatic symptom disorder patients. This difference remained significant in the subgroup of MRZ-negative RRMS patients. In conclusion, the measurement of NOx in the serum might indeed be a valuable tool in supporting MS diagnosis.

## 1. Introduction

Nitric oxide (NO) is a free radical which plays a pivotal role in health and disease [1]. NO is produced by the enzyme NO synthase (NOS), which catalyzes the synthesis of NO and L-citrulline from L-arginine. There are three isoforms of the NOS: neuronal NOS (nNOS), inducible NOS (iNOS), and endothelial NOS (eNOS). NNOS and eNOS are constitutively expressed in neuronal and endothelial cells, respectively. INOS is present at low levels under physiological conditions but its expression is also induced by different proinflammatory cytokines such as tumor necrosis factor α (TNF-α) and interleukin-1β (IL-1β) [2,3]. Even though NO is critical for many physiological processes, high concentrations can lead to cell damage [4]. This is based on NO reacting with O_2_^−^, which leads to the formation of the toxic metabolite peroxynitrite (ONOO^−^). ONOO^−^ is in equilibrium with peroxynitrous acid (ONOOH), both of which can directly react with different biomolecules. ONOOH can also form one-electron oxidant hydroxyls (OH) and NO_2_ radicals. This results in lipid oxidation and protein nitration yielding; for instance, 3-nitrotyrosine (3-NT). Furthermore, ONOO^−^ can react with carbon dioxide (CO_2_), resulting in nitrogen dioxide (NO_2_) radicals and carbonate (CO_3_^−^), which in turn mediate the oxidation and nitration of proteins and DNA [5]. Apart from the toxic effects of peroxynitrite, high levels of NO can also impair glycolysis as well as oxidative phosphorylation and cause mitochondrial damage via the S-nitrosylation of proteins [4,6,7].

MS is an immune-mediated disorder which affects different parts of the CNS. Pathophysiologically, inflammatory attacks mediated by autoreactive innate and adaptive immune cells together with a dysregulated humoral immune response contribute to demyelination and axonal damage [8,9]. There is increasing evidence that NO produced by immune or CNS-resident cells is involved in the pathogenesis of MS [10]. Mechanistically, the release of proinflammatory cytokines induces iNOS expression in reactive microglia, astrocytes, and macrophages, which contributes to the synthesis of NO [1]. NO mediates vasodilation and contributes to the disruption of the blood–brain barrier (BBB), facilitating the entry of substances and the transmigration of leukocytes into the CNS [11]. In addition, NO increases the adherence of leukocytes to the blood vessel endothelium and enhances diapedesis [4]. However, NO also exerts protective effects, which could be shown in the experimental autoimmune encephalomyelitis (EAE) model of MS [12]. Thus, the potential therapeutic targeting of nitrosative molecules in MS should be selective in order to prevent harmful effects of NO while preserving beneficial physiological actions.

The clinical phenotype of MS can be divided into three subtypes: relapsing–remitting MS (RRMS), primary progressive MS (PPMS), and secondary progressive MS (SPMS) [13]. Pathophysiological mechanisms differ depending on the MS phenotype. For instance, the activation of the peripheral immune response, which targets the CNS, drives the relapsing–remitting disease course. In this context, the infiltration of peripheral immune cells leads to the disruption of the BBB, primarily in the white matter of the CNS, resulting in the formation of classical active demyelinating lesions [14,15]. In the progressive forms of MS, a combination of immune effector mechanisms (mainly within the CNS), neurodegeneration, and axonal dysfunction is involved in the disease pathology [14]. While BBB damage is thought to play a subordinate role in progressive forms of MS, the activation of and damage to CNS resident cells is considered to be crucial. In this regard, NO production by astrocytes has been identified as being implicated in cytokine-induced neurotoxicity [16,17].

Previous studies have assessed the potential of nitrosative stress molecules (NOx) as biomarkers for MS [18,19,20,21,22,23,24,25]. We recently published a meta-analysis of 22 studies which found elevated levels of NOx in the cerebrospinal fluid (CSF) of MS patients compared to patients with noninflammatory other neurological diseases (NIOND). Serum levels, however, were not significantly different between those two groups. In comparison to healthy controls (HC), no statistically significant differences of NOx levels in serum or CSF could be detected in MS patients. However, many of the included studies did not distinguish between the different clinical subtypes of MS, included pre-treated patients, and inclusion criteria for the NIOND and the HC groups varied [10]. In this follow-up study, we therefore aimed to analyze the serum and CSF NOx levels in clinically well-defined cohorts of treatment-naïve patients with RRMS and PPMS compared to patients with somatic symptom disorder (Soma) to assess the potential of NOx as biomarkers facilitating the diagnosis of MS.

## 2. Results

### 2.1. Basic Demographics and Disease Characteristics

In total, 117 patients (71 RRMS, 16 PPMS, and 30 Soma) were eventually included in the study. Demographics and basic disease characteristics are shown in Table 1 and Appendix A.

### 2.2. Elevated NOx in the Serum of RRMS Patients Compared to Soma

A comparison between RRMS and Soma patients revealed significantly elevated levels of NOx in the serum of RRMS patients (Figure 1A). In accordance with the literature, the majority of RRMS patients had positive oligoclonal bands (OCBs) in the CSF [26]. Apart from OCBs, which can serve as proof of dissemination in time according to the revised 2017 McDonald diagnostic criteria for MS [27], the MRZ reaction (antibody indices (AI) against measles, rubella, and varicella zoster virus) is another parameter supporting the diagnosis of RRMS. While many RRMS patients show a positive MRZ reaction (defined as at least two out of three AI higher than 1.5), negative results can be found in many cases of other autoimmune inflammatory neurological diseases and infectious CNS diseases [28]. Conversely, the diagnosis of RRMS can be challenging in patients with negative OCBs and a negative MRZ reaction. We therefore aimed to assess the diagnostic value of NOx in this patient subgroup. There was only a trend towards increased NOx levels in the serum of RRMS patients with negative CSF OCBs (Appendix A). However, RRMS patients with a negative MRZ reaction featured significantly elevated NOx levels in the serum (Figure 1B). Of note, even when only RRMS patients with double-negative OCBs and MRZ reaction were compared to Soma patients, the difference in serum NOx remained significant (Figure 1B).

As NOx seems to be involved in acute as well as chronic neuroinflammation, we also assessed whether differences in serum NOx levels could be detected between RRMS patients with and without contrast enhancing lesions on cranial magnetic resonance imaging (cMRI), as well as patients with and without acute relapse. In this context, no relevant differences were found, respectively (Figure 1C,D). In order to assess potential age and sex specific differences, we compared serum NOx levels between female and male RRMS patients and performed simple linear regressions with the ages of the RRMS patients. No relevant effects of either sex or age were detected (Figure 1E,F). Furthermore, we correlated serum NOx levels with different clinical and paraclinical parameters. Using simple linear regression, we found no significant association between serum NOx levels and Expanded Disability Status Scale (EDSS) at the time of baseline sampling and disease duration (Figure 1G and Appendix A). However, a positive relationship was found between serum NOx levels and the CSF/serum albumin ratio (QAlb) as a marker of blood–CSF barrier (BCSFB) dysfunction (Figure 1H). In line with this, elevated NOx serum levels were associated with higher NOx in the CSF (Figure 1I). Routine CSF analysis was performed ± three months of NOx analysis. As serum NOx levels were highly variable within the RRMS cohort, we investigated if there were differences in clinical parameters between RRMS patients with high NOx levels (defined as higher than the upper limit of Soma NOx) and patients with low serum NOx levels. Comparisons of relapses, active lesions on cMRI, disease durations, and EDSSs at baseline sampling did not reveal significant differences of NOx levels between the two groups (Figure 1J and Appendix A). Nine RRMS patients featured very high NOx levels in the serum of more than 20 µM. For two of these patients, clinical follow-up data were available. One was started on diroximel fumarate and the other on ofatumumab. Both had either signs of clinical or paraclinical disease activity within the first year of treatment. Taken together, RRMS patients featured higher serum NOx levels in comparison to Soma patients. In addition, serum NOx levels positively correlated with the QAlb in this cohort. In the subgroup of MRZ-negative RRMS patients, the difference in serum NOx compared to Soma remained significant.

### 2.3. No Relevant Differences in CSF NOx Levels between RRMS and Soma Patients

Our previous meta-analysis found significantly increased CSF NOx levels in RRMS patients compared to NIOND [10]. We therefore measured CSF NOx levels in 26 RRMS compared to 12 Soma patients, but did not find significant differences between the two groups (Figure 2A). Similar to serum, CSF NOx levels were comparable between female and male RRMS patients, and no relevant effect of the patients’ ages on CSF NOx levels was observed (Figure 2B,C). Furthermore, CSF NOx levels did not significantly differ between relapse and remission and between patients with or without active cMRI lesions (Appendix A). Nevertheless, there was a positive association between the EDSS and CSF NOx levels (Figure 2D), while no relevant impact was found for QAlb, disease duration, and different CSF routine parameters at baseline sampling (Figure 2E and Appendix A). In summary, CSF NOx levels were comparable between RRMS and Soma patients and were not influenced by clinical or paraclinical parameters, with the exception of the EDSS.

### 2.4. PPMS Patients Have Increased Serum NOx Levels Compared to Soma but Not to RRMS Patients

As data on NOx in PPMS are overall scarce and only a few studies included PPMS patients as a separate MS cohort, we repeated our analysis of NOx levels using serum from treatment-naïve PPMS patients. PPMS patients featured significantly higher serum NOx levels in comparison to Soma patients, whereas no significant differences were detected between the PPMS and the RRMS cohort (Figure 3A,B). In parallel to the RRMS cohort, no age- and sex-related differences of serum NOx serum levels were found in PPMS patients (Figure 3C,D). Furthermore, we did not detect a relevant impact of disease duration, EDSS or QAlb at baseline sampling on serum NOx levels (Appendix A). Unfortunately, CSF was only available from one PPMS patient. Thus, no analysis of NOx in the CSF could be performed for this cohort. In summary, higher levels of NOx were detected in the serum of PPMS patients compared to Soma but not in comparison to RRMS patients.

## 3. Materials and Methods

### 3.1. Study Population and the Collection of Serum and Cerebrospinal Fluid (CSF) Samples

Patients presenting with symptoms suggesting the differential diagnosis of RRMS and PPMS, who were admitted to the Department of Neurology of the Heinrich Heine University (HHU) Düsseldorf between 17 February 2021 and 30 September 2023, and who did not meet the exclusion criteria, were included in the study.

Patients were excluded based on the following criteria:Treatment with disease modifying therapies (DMTs), immunosuppressants, steroids, or immunoadsorption/plasmapheresis at time of baseline sampling;Previous treatment with DMTs, immunosuppressants, total body irradiation, or bone marrow transplantation;Treatment with steroids or immunoadsorption/plasmapheresis within the last 4 weeks prior to sampling;Medical, psychiatric, cognitive, or other conditions compromising the patient’s ability to give informed consent;Patients with other systemic inflammatory diseases (e.g., SLE, rheumatoid arthritis, vasculitis);Patients with known active malignancies;Patients with known immunodeficiencies;Patients with acute or chronic infectious diseases (e.g., infection with the human immunodeficiency viruses (HIV), hepatitis B or C, tuberculosis).

Serum and CSF were collected during clinical routine workup. BD Vacutainer^®^ SST II Advance Tubes and BD Vacutainer^®^ Serum Tubes (Becton Dickinson, Franklin Lakes, NJ, USA) were used, respectively. Centrifugation (1500× *g*, 20 °C, 10 min) of the blood was performed after 30 min at room temperature. The serum was transferred into screw-cap micro tubes (Starstedt, Nümbrecht, Germany) and stored at −20 °C until further analysis. The CSF was centrifuged at 250× *g* and 4 °C for 25 min. Subsequently, the supernatant was collected, transferred to screw-cap micro tubes (Starstedt, Nümbrecht, Germany) and stored at −20 °C. The study was performed according to the Declaration of Helsinki and was approved by the local Ethics Committee of the Board of Physicians of the Region Nordrhein and of the Heinrich Heine University Düsseldorf, Germany (reference number: 5951R). All patients gave written informed consent to participate in the study.

### 3.2. The Retrospective Identification of Patients

We retrospectively queried the clinical database of the Department of Neurology of the HHU Düsseldorf to identify patients with available serum and/or CSF samples who had received the diagnosis of RRMS or PPMS according to the 2017 revised McDonald criteria [27], or Soma based on the ICD-10 diagnostic criteria [29]. Initially, 169 patients were identified. Fifty-two of these had to be excluded due to diagnostic uncertainty, alternative diagnosis, incomplete diagnostic workup or one of the above-mentioned exclusion criteria. Thus, 117 patients (71 RRMS, 16 PPMS, and 30 Soma) were eventually included in the study. For the RRMS and PPMS cohorts, the following clinical parameters were assessed at baseline sampling: disease duration, comorbidities, medication, routine CSF parameters, acute symptoms meeting the criteria of a clinical relapse, EDSS, and presence of active lesion on MRI. For the Soma patients, comorbidities, medication, and routine CSF parameters were assessed. The study design is illustrated in Figure 4.

### 3.3. The Analysis of Nitric Oxide Metabolites

Given the short half-life of NO, measurements of the more stable NO metabolites nitrite (NO_2_^−^) and nitrate (NO_3_^−^) are used to indirectly measure NO levels in biological fluids [2]. The Nitric Oxide Assay Kit (Colorimetric) ab65328 (abcam, Cambridge, UK) was used for the detection of NO metabolites according to the manufacturer’s instructions. In brief, nitrate reductase is used to convert nitrate to nitrite. Next, the Griess Reagent was applied to convert nitrite to a purple azo compound. The amount of the azochromophore was measured with a microplate reader and reflected the amount of NO in the samples.

### 3.4. Statistical Analysis

R studio and ‘GraphPad Prism’ (version 9.0.0) were used for data analysis and visualization. As normality could not be assumed based on the D’Agostino and Pearson test, Mann–Whitney U test was used for comparison of two groups, and Kruskal–Wallis test with Dunn test were used to correct for multiple comparisons for more than two groups. For binary data, Fisher’s exact test was applied. Simple linear regression was used to analyze the association between NOx in serum or CSF and different clinical parameters. A *p*-value of ≤0.05 was considered significant.

## 4. Discussion

This study investigates the intricate role of NO in MS and explores the value of NO metabolites nitrate/nitrite (NOx) as potential biomarkers for diagnosing MS and to improve the understanding of the disease. The overarching objective of our study was to investigate the role of NOx as a diagnostic tool to differentiate MS from patients presenting with symptoms indicative of MS, but without further signs of an inflammatory CNS disease.

In general, the growing focus on biomarker discovery in MS represents a new approach to the diagnosis and treatment of the disease. Biomarkers hold promise not only for enhancing diagnostic precision but also for prognosticating disease progression and evaluating treatment response. This study builds on our prior meta-analysis investigating the diagnostic value of NOx in the serum and CSF of patients with symptoms suspicious of MS. In this meta-analysis, we found heterogeneous and sometimes even contradictory results, mostly owing to inconsistent inclusion and exclusion criteria and other factors. In the present study, we therefore aimed to provide an enhanced assessment of the value of NOx by employing stringent inclusion and exclusion criteria in order to increase the precision and reliability of our findings. The key finding of this study is that in the serum of MS patients NOx levels are significantly increased compared to somatic symptom disorder patients. This underlines the diagnostic significance of NOx in discerning patients with MS from those who present with clinical symptoms suspicious of MS but without further signs of a chronic inflammatory CNS disease. Given the scope of our study, we did not include a ‘healthy’ control group, as the clinical reality physicians are facing is that patients seek an explanation for their physical symptoms—irrespective of their etiology. Based on our results, one could assume that serum NOx levels are higher in MS compared to healthy controls; however, this could not be confirmed by our previous meta-analysis. Thus, future studies will be necessary for clarification. In general, our results confirm previous studies which also suggested an increased level of nitrosative stress in MS patients with the important restriction that they were performed in smaller patient groups and with inconsistent control groups. Notably, our findings are irrespective of age and sex, which we could rule out as potential confounders in our cohort. Interestingly, our study identified a subgroup of RRMS patients with elevated NOx levels in the serum but negative OCBs and a negative MRZ reaction in the CSF. This highlights the potential of NOx as a biomarker in situations where, despite using the current diagnostic standard, the diagnosis remains unclear. Interestingly, we found no significant differences in NOx levels related to clinical and paraclinical disease activity (relapses and MRI active lesions) which, at first sight, seems counter-intuitive. However, it is conceivable that in contrast to neurofilament (NF), which was found to be elevated as a result of increased (inflammatory) neuronal injury, the kinetics of NOx in MS are more complex. This certainly warrants future longitudinal approaches. Additionally, and as expected, the correlation of serum NOx levels with CSF/serum QAlb suggests a link between NO and BCSFB dysfunction, further implicating NOx as a potential marker for the MS-related breakdown of BCSFB integrity. In addition, higher NOx serum levels were associated with elevated NOx in the CSF. Taking those findings together, one could speculate that NOx synthesized in the periphery enters the CSF via the impaired BCSFB. Nevertheless, in contrast to the serum, CSF NOx levels did not significantly differ between RRMS and Soma patients, indicating that the diagnostic value of NOx is more pronounced in the serum than in the CSF. Notably, even though the investigated groups are only partly comparable, this is a not in line with the results of our meta-analysis [10]. Pathophysiologically, however, this might simply point to the pivotal contribution of the peripheral immune system to the inflammatory processes in MS. On the other hand, we found a positive correlation between the EDSS and overall CSF NOx levels. For two RRMS patients featuring very high NOx levels in the serum (more than 20 µM), clinical follow-up data were available. Both patients had signs of clinical and/or paraclinical disease activity within the first year of treatment, indicating the potential of NOx as a prognostic biomarker in MS. Taken together, this underlines that, irrespective of its origin, there is a link between NOx levels and the clinical course of MS. Nevertheless, further studies with larger sample sizes will be needed to address this aspect.

Furthermore, our study also focused on PPMS patients, revealing higher serum NOx levels compared to Soma patients, while there was no significant difference between this MS subtype and RRMS. This demonstrates a contribution of NOx to the pathological processes in both disease variations, suggesting that nitrosative stress molecules increase both neuroinflammation and neurodegeneration.

We acknowledge that one of our study’s limitations is the unavailability of CSF data for the majority of the PPMS cohort, and the lack of a control group with other inflammatory CNS diseases such as, for instance, neuromyelitis optica spectrum disorders (NMOSD) and autoimmune encephalitis (AIE). Since both shortcomings are a result of a lower prevalence compared to RRMS, register studies would be an adequate future tool to address this issue. In general, longitudinal study approaches in diseases such as NMOSD, AIE, and other disease entities could provide a deeper insight into the mechanics and kinetics of NOx in CNS diseases. The exact duration of such studies remains, however, unclear, due to the diverse pathogeneses of the diseases in question. In this respect, specificity might certainly be an issue regarding the measurement of NOx. Nevertheless, NOx might prove useful as a ‘screening parameter’ to identify patients benefitting from a thorough diagnostic workup. For example, with regard to the differential diagnoses of NMOSD and AIE, subsequent testing for anti-AQP4 antibodies and other autoantibodies can provide enhanced diagnostical safety. What is more, a combination of NOx with other biomarkers might increase sensitivity in the future. We also did not include patients with secondary progressive MS (SPMS), as this entity has predominantly received prior treatment, and therefore fell under our exclusion criteria. Having said that, it is obvious that NOx in SPMS would be particularly relevant as a progression and therapy response marker, which was not the focus of our study.

In conclusion, this study confirms the value of NOx as a diagnostic tool for RRMS in the serum. However, further investigations in larger cohorts and with a longitudinal design will be needed to fully uncover the diagnostic and prognostic potential of this group of molecules in the context of MS.

## Figures and Tables

**Figure 1 ijms-25-00787-f001:**
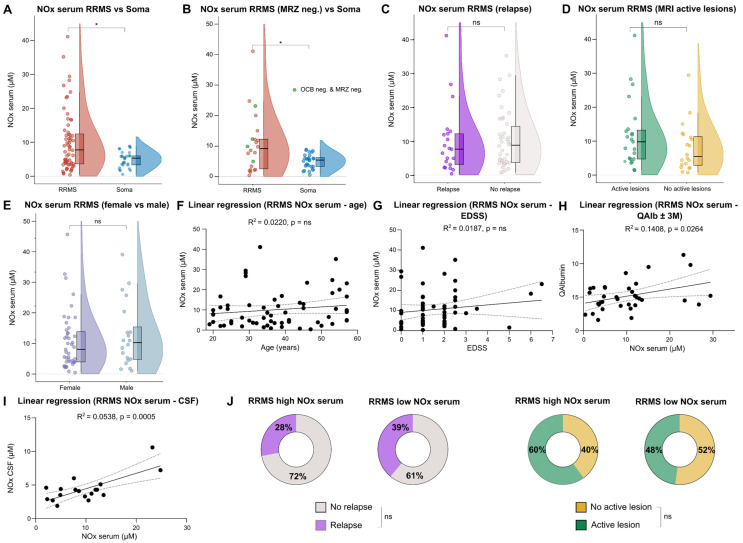
RRMS patients feature elevated NOx serum levels compared to somatic symptom disorder patients (Soma). (**A**–**E**) Violin plots with overlaying box plots comparing two groups. Boxes display the median as well as the 25th and 75th percentiles. The whiskers extend from the hinge to the largest and smallest values, respectively, but no further than 1.5 * IQR from the hinge. (**F**,**G**) Simple linear regression using either age or EDSS as independent and NOx serum as dependent variable. The area in-between the dotted lines shows the 95% confidence band. (**H**,**I**) Simple linear regression using NOx serum as independent and either QAlbumin or NOx CSF as dependent variable. The area in-between the dotted lines shows the 95% confidence band. (**J**) Pie charts comparing the percentage of patients with acute relapse and active lesions on cMRI at time of sampling between the high and low serum NOx group. CSF—Cerebrospinal fluid, EDSS—Expanded Disability Status Scale, IQR—Interquartile range, M—Months, MRI—Magnetic resonance imaging, MRZ—Antibody indices (AI) against measles, rubella, and varicella zoster virus. MRZ was defined ‘positive’ if at least two out of three AI were higher than 1.5, neg—Negative, NOx—Nitrosative stress molecules, ns—Not significant, OCBs—Oligoclonal bands, Q—CSF/serum ratio, RRMS—Relapsing–remitting Multiple Sclerosis, Soma—Somatic symptom disorder, *—multiplied by.

**Figure 2 ijms-25-00787-f002:**
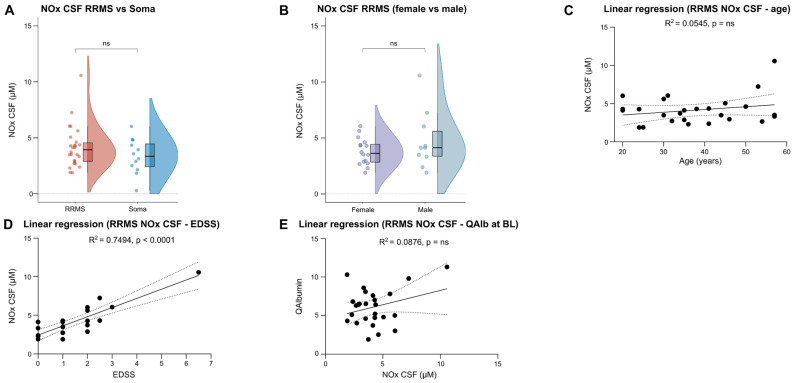
CSF NOx levels do not differ significantly between RRMS and Soma patients. (**A**,**B**) Violin plots with overlaying box plots comparing two groups. Boxes display the median as well as the 25th and 75th percentiles. The whiskers extend from the hinge to the largest and smallest values, respectively, but no further than 1.5 * IQR from the hinge. (**C**,**D**) Simple linear regression using either age or EDSS as independent and NOx CSF as dependent variable. The area in-between the dotted lines shows the 95% confidence band. (**E**) Simple linear regression using NOx CSF as independent and QAlbumin as dependent variable. The area in-between the dotted lines shows the 95% confidence band. BL—Baseline, CSF—Cerebrospinal fluid, EDSS—Expanded Disability Status Scale, IQR—Interquartile range, NOx—Nitrosative stress molecules, ns—Not significant, Q—CSF/serum ratio, RRMS—Relapsing–remitting Multiple Sclerosis, Soma—Somatic symptom disorder.

**Figure 3 ijms-25-00787-f003:**
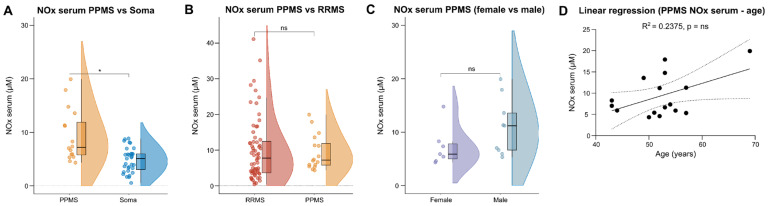
Increased nitrosative stress in serum of PPMS patients compared to Soma patients. (**A**–**C**) Violin plots with overlaying box plots comparing two groups. Boxes display the median as well as the 25th and 75th percentiles. The whiskers extend from the hinge to the largest and smallest values, respectively, but no further than 1.5 * IQR from the hinge. (**D**) Simple linear regression using age as independent and NOx serum as dependent variable. The area in-between the dotted lines shows the 95% confidence band. IQR—Interquartile range, NOx—Nitrosative stress molecules, ns—Not significant, PPMS—Primary progressive Multiple Sclerosis, Q—CSF/serum ratio, RRMS—Relapsing–remitting Multiple Sclerosis, Soma—Somatic symptom disorder, *—multiplied by.

**Figure 4 ijms-25-00787-f004:**
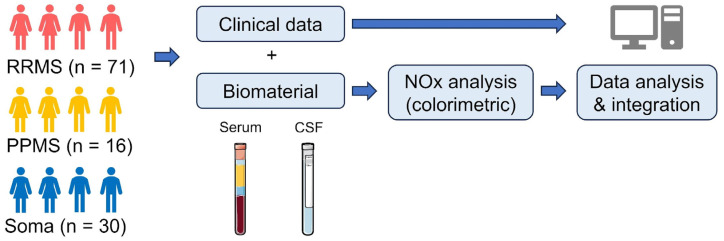
Study design. CSF—Cerebrospinal fluid, NOx—Nitrosative stress molecules, PPMS—Primary progressive Multiple Sclerosis, RRMS—Relapsing–remitting Multiple Sclerosis, Soma—Somatic symptom disorder. Parts of the figure were drawn by using pictures from Servier Medical Art (https://smart.servier.com/, accessed on 1 September 2023). Servier Medical Art by Servier is licensed under a Creative Commons Attribution 3.0 Unported License (https://creativecommons.org/licenses/by/3.0/, accessed on 1 September 2023).

**Table 1 ijms-25-00787-t001:** Basic demographics and disease characteristics of the RRMS, PPMS, and Soma cohorts.

	RRMS	PPMS	Soma
Total no. patients	71	16	30
No. patients with serum samples	63	16	28
No. patients with CSF samples	26	1	12
No. patients with serum and CSF samples	18	1	10
Age (median [range])	36 [19–57]	52 [43–70]	29 [19–55]
Sex (% female)	67.61	43.75	86.67
Disease duration (median [range]) (Y)	0.66 [0.00–19.73]	4.76 [0.97–27.38]	n/a
CSF pleocytosis (% of patients)	68.97	64.29	0
BCSFBD (% of patients)	39.22	55.56	14.81
Positive OCBs (% of patients)	88.06	92.86	0
Positive MRZ (% of patients)	44.44	55.56	0
Intrathecal IgG synthesis (% of patients)	85.97	92.86	0
Intrathecal IgM synthesis (% of patients)	21.74	20.00	0
Intrathecal IgA synthesis (% of patients)	17.39	0	0

BCSFBD—Blood–CSF-barrier dysfunction, CSF—Cerebrospinal fluid, Ig—Immunoglobulin, MRZ—Antibody indices (AI) against measles, rubella, and varicella zoster virus. MRZ was defined ‘positive’ if at least two out of three AI were higher than 1.5, No.—Number of, OCBs—Oligoclonal bands, PPMS—Primary progressive Multiple Sclerosis, RRMS—Relapsing–remitting Multiple Sclerosis, Soma—Somatic symptom disorder, Y—Years.

## Data Availability

Further information and requests for resources and anonymized clinical data should be directed to and will be fulfilled by David Kremer (david.kremer@artemed.de).

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
