# Peer review of "The Use of Nitrosative Stress Molecules as Potential Diagnostic Biomarkers in Multiple Sclerosis"

_ijms, 2024, doi:10.3390/ijms25020787_

Round 1
Reviewer 1 Report
Comments and Suggestions for Authors
Multiple sclerosis (MS) is an autoimmune disease affecting the central nervous system. This study conducted a meta-analysis to investigate the nitrite/nitrate (NOx) levels in MS patients compared with non-inflammatory neurological diseases. Both cerebrospinal fluid (CSF) and serum data were studied. This study suggests that serum NOx levels can be a tool for the diagnosis of relapsing-remitting primary progressive MS. I have several comments.
1. The Introduction section can be separated into several paragraphs.
2. Line 165. Change “Table 1-6” to “Tables 1-6.”
3. In Table 1, how many patients have both CSF samples and serum samples?
For the MS patients with both samples, does the level of NOx consistently increase or decrease compared with the Soma group?
4. Figure 2c shows no significant difference in the NOx level between relapse and no relapse groups. However, in Figure 2I, the percentages of 28% and 72% appear to have a significant difference. This figure may confuse readers. It seems that both high and low NOx levels are no relapse patients. What are the sample sizes of relapse and no relapse groups?
5. This study concluded that RRMS and PPMS patients had higher serum NOx levels compared to somatic symptom disorder patients. In general, do MS have higher or lower NOx levels compared to healthy individuals? You may have some discussions on it.
Comments on the Quality of English LanguageModerate editing of the English language is required.
Reviewer 2 Report
Comments and Suggestions for Authors_The introduction of the paper is comprehensive in terms of technical information about nitric oxide (NO) and its role in multiple sclerosis (MS), but there are areas that could be considered for improvement:
The structure of the introduction text could be enhanced:
-Some sentences and phrases could be simplified or divided to improve clarity and readability.
-Despite providing a lot of detailed information, the introduction could benefit from a better connection between the presented ideas and their direct relevance to the main topic. The relationship between NO and the different subtypes of MS is not clearly articulated. Overall, simplifying and organizing the information in a clearer manner, while maintaining focus on the current study's objectives, could enhance accessibility and general comprehension of the introduction.
_Materials and methods are very well written and grounded.
_Regarding the discussion, despite the abundance of information, at certain points, the discussion may seem somewhat scattered.
It would be beneficial to summarize some key findings and maintain a more specific focus on the study's objectives and results.
Limitations are mentioned, such as the lack of data in certain subgroups or the absence of certain controls.
It would be useful to delve deeper into how these limitations might have affected the results or the interpretation of the study.
It would be advisable to highlight more clearly how the obtained results relate to previous studies. Do they confirm or contradict previous findings? This connection could provide more context to the current findings.
Future research needs are mentioned, which is positive. However, they could elaborate a bit more on what types of studies would be necessary and how identified limitations could be addressed.
Other than that, it's a great research effort.
